# Bojungikgi-Tang, a Traditional Herbal Formula, Exerts Neuroprotective Effects and Ameliorates Memory Impairments in Alzheimer’s Disease-Like Experimental Models

**DOI:** 10.3390/nu10121952

**Published:** 2018-12-10

**Authors:** Hye-Sun Lim, Yu Jin Kim, Eunjin Sohn, Jiyeon Yoon, Bu-Yeo Kim, Soo-Jin Jeong

**Affiliations:** 1Clinical Medicine Division, Korea Institute of Oriental Medicine, Daejeon 34054, Korea; qp1015@kiom.re.kr (H.-S.L.); jinjin0228@kiom.re.kr (Y.J.K.); ssen4022@kiom.re.kr (E.S.), jyeon7139@kiom.re.kr (J.Y.); buykim@kiom.re.kr (B.-Y.K.); 2Herbal Medicine Resources Research Center, Korea Institute of Oriental Medicine, Jeollanam-do 58245, Korea; 3College of Pharmacy, Chungnam National University, Daejeon 34134, Korea

**Keywords:** Alzheimer’s disease, amyloid-β, Bojungikgi-tang, memory loss, neuroprotection

## Abstract

Bojungikgi-tang (BJIGT; Bu Zhong Yi Qi Tang in China, Hochuekkito in Japan) is a traditional Oriental herbal formula comprised of eight medicinal herbs that has long been used for the treatment of digestive disorders. A recent clinical study from South Korea reported that BJIGT-gamibang administration may be effective in treating dementia. We aimed to establish scientific evidence for the anti-dementia effects of BJIGT using in vitro and in vivo experimental models. We measured amyloid- β (Aβ) aggregation, β-secretase (BACE), and antioxidant activity in a cell free system. Neuroprotective effects were assessed using CCK-8. Imprinting control region (ICR) mice were divided into the following six groups: Normal control, Aβ-injected, Aβ-injection + oral BJIGT gavage (200, 400, or 800 mg/kg/day), and Aβ-injection + oral morin administration (10 mg/kg/day). Subsequently, behavioral evaluations were conducted and brain samples were collected from all the animals and assessed. BJIGT enhanced inhibition of Aβ aggregation and BACE activity in vivo, as well as antioxidant activity in in vitro, cell-free systems. BJIGT also exerted neuroprotective effects in a hydroperoxide (H_2_O_2_)-induced damaged HT22 hippocampal cell line model. In addition, BJIGT administration significantly ameliorated cognitive impairments in Aβ-injected mice, as assessed by the passive avoidance and Y-maze tests. Furthermore, BJIGT treatment suppressed Aβ aggregation and expression, as well as expression of Aβ, NeuN, and brain-derived neurotrophic factor (BDNF) in the hippocampi of Aβ-injected mice. Overall, our results demonstrate that, with further testing in clinical populations, BJIGT may have great utility for the treatment of dementia and especially Alzheimer’s disease.

## 1. Introduction

Dementia is an umbrella term used to describe a wide range of mental pathologies, caused by damage to the brain. Symptoms of dementia include memory loss and language and behavior problems. If severe, dementia eventually leads to death. Alzheimer’s disease (AD) is the most common type of dementia and has several known etiological contributors, including genetics, lifestyle, and environmental factors, although its primary cause remains poorly understood. The second most frequent form of dementia is vascular dementia (VD), also known as vascular cognitive impairment, which occurs as a result of decreased blood supply to the brain [1]. In total, AD and VD make up 80–90% of total dementia cases, and the number of individuals with these diagnoses is gradually increasing. Despite this, no curative medical treatment has been developed or approved by the U.S. Food and Drug Administration (FDA) for use in these diseases, and at present only symptom relief medications exist [2]. Therefore, dementia drug development is an area of vast medical unmet need.

Bojunikgi-tang (BJIGT), also known as Hochuekkito in Japan or Bu Zhong Yi Qi Tang in China, is a well-known traditional herbal formula that ranks fifth in terms of cost among the 56 herbal formulas covered by National Health Insurance in South Korea. BJIGT is traditionally prescribed to treat gastric disorders such as gastroptosis, leiasthenia, and myasthenia gastrica [3]. Studies have also reported that BJIGT may act against liver, lung, and gastric cancer development [4,5,6], as well as synergistically with chemotherapeutics such as cisplatin [7,8]. BJIGT is also known to be effective in treating multiple inflammatory, neurological, and respiratory diseases [9,10,11,12]. Of particular interest, recent clinical reports have demonstrated that BJIGT-gamibang improves symptoms in patients with dementia [13,14]. However, these reports focused on vascular dementia only, and additional scientific questions remain with regard to the mechanisms which underlie BJIGT’s efficacy in treating dementia. In the present study, we explored the effects of BJIGT and mechanisms which may underlie these effects using in vitro and in vivo AD-like models.

## 2. Materials and Methods

### 2.1. BJIGT and Reagents

BJIGT was obtained from Han Kook Shin Yak Pharmaceutical Co., Ltd. (Nonsan, Chungnam, Korea) and dissolved in phosphate buffered saline (PBS; HyClone Laboratories Inc., Logan, UT, USA) for use in both in vivo and in vitro experiments. A voucher specimen (SCD-B-008) has been deposited at the Clinical Medicine Division, Korea Institute of Oriental Medicine.

Liquiritin apioside, liquiritin, nodakenin, and decursin were purchased from Sunny Biotech Co., Ltd. (Shanghai, China). Hesperidin and decursinol angelate were purchased from Biopurify Phytochemicals (Chengdu, China). Glycyrrhizin was purchased from ChemFaces Biochemical Co., Ltd. (Wuhan, China). The purities of these compounds were all assessed to be ≥98.0% by high-performance liquid chromatography (HPLC) analysis. HPLC-grade solvents, acetonitrile, water, analytical grade reagents, and acetic acid were purchased from J. T. Baker Chemical Co. (Phillipsburg, NJ, USA).

### 2.2. Apparatus and Chromatographic Conditions

Quantitative analyses were conducted using a Waters Alliance e2695 system (Waters Corp., Milford, MA, USA) equipped with a pump, degasser, column oven, autosampler, and photodiode array detector (Waters Corp., #2998). These data were acquired and processed using Empower software (version 3; Waters Corp). Chromatographic separation of the seven standard components of BJIGT was carried out at room temperature using Gemini C_18_ analytical columns (250 × 4.6 mm, 5 μm) from Phenomenex (Torrance, CA, USA), with a gradient solvent system of 1.0% (v/v) aqueous acetic acid (A) and acetonitrile (B). The elution conditions were as follows: 12–60% B for 0–30 min, 60–65% B for 30–40 min, 65–100% B for 40–45 min, and 100% B for 45–60 min. The flow rate was 1.0 mL/min and the injection volume was 10 μL. The ultraviolet (UV) wavelength range of the PDA detector was 190 nm to 400 nm.

### 2.3. Preparation of Standard Solution

The seven components of BJIGT were weighed accurately, dissolved in methanol at 1.0 mg/mL, and stored below 4 °C. Stock solutions were further diluted to yield a series of standard dilution solutions of different concentrations for quantitative analyses.

### 2.4. Preparation of Sample Solutions

The BJIGT formulation was weighed and dissolved in 10% aqueous DMSO at 20 mg/mL. This solution was then filtered through a syringe filter (0.45 μm) for HPLC analysis.

### 2.5. Calibration Curve and Determination of the Limit of Detection (LOD) and Quantification (LOQ)

The calibration curves of BJIGT components were calculated from the peak areas of the standard solutions at different concentrations. The tested concentration ranges were as follows: For liquiritin apioside and hesperidin, 1.56–100 μg/mL; for liquiritin and decursinol angelate, 0.39–25 μg/mL; for nodakenin and decursin, 0.78–50 μg/mL; and for glycyrrhizin, 6.25–400 μg/mL. These solutions were measured in triplicate to establish each calibration curve. The LOD and LOQ for the seven standard components were calculated using the slope of the calibration curve and the standard deviation (SD) of the intercept as follows: LOD = 3.3 × (SD of the response/slope of the calibration curve); and LOQ = 10 × (SD of the response/slope of the calibration curve).

### 2.6. Aβ Aggregation Assay

Aβ aggregation was measured using the SensoLyte^®^ Thioflavin T β-Amyloid aggregation kit (AnaSpec, Inc., Fremont, CA, USA), according to the manufacturer’s instruction. The assay is based on the properties of thioflavin T dye, in which fluorescence is increased when bound to aggregates of Aβ_1-42_ peptides. Briefly, Thioflavin T was dissolved in assay buffer [50 mM Tris/150 mM NaCl (pH = 7.2), 20 mM HEPES/150 mM NaCl (pH = 7.2), 10 mM phosphate/150 mM NaCl] and used at a concentration of 100 μM. Samples were dissolved in assay buffer and used at a final concentration of 100 μg/mL. To measure the inhibition of Aβ aggregation, 5 μL of the sample, 85 μL of Aβ, and 10 μL of thioflavin T were mixed in each well of a black, 96-well microplate. Thioflavin T fluorescence was measured at intervals of 20 min for 2 h with an excitation wavelength (λ_ex_) of 440 nm and an emission wavelength (λ_em_) of 485 nm using the SpectraMax i3 Multi-Mode Detection Platform (Molecular devices, LLC., CA, USA). This yielded seven readings from each sample/well. All fluorescence readings were expressed in relative fluorescence units. Morin was used as a positive control. Experiments were performed in triplicate and averaged, with the percentage inhibition of Aβ aggregation calculated according to the following equation: Aβ aggregation inhibition (%) = (1-Fluorescence of Aβ − treated sample)/(Fluorescence of untreated sample) × 100.

### 2.7. Cell-Free BACE Activity Assay

The Cell free BACE activity assay was performed according to the manufacture’s protocol using a commercially available SensoLyte^®^ 520 BACE assay kit (Anaspec Inc, Fremont, CA, USA) at a final volume of 100 μL in a black 96-well microplate. Serially-diluted BJIGT extract samples at a concentration of 100 mg/mL were prepared by dilution in assay buffer to final concentrations of 100, 50, and 25 μg/mL. For enzymatic reactions in a microplate, 40 μL of human μ-secretase enzyme solution and 10 μL of each dilution of BJIGT extract sample (one concentration per reaction) were mixed for 10 min at room temperature. To measure cell-free BACE activity assay with inhibitors, 10 μL of inhibitor solution was added to make a 0.25 μM final concentration in each microplate well. The 50 μL of BACE substrate solution was then added to the mixture to start a reaction in each independent well. The reagents were mixed by shaking the plate gently for 30 sec. Fluorescent signal was measured by a fluorescence microplate reader (Spectra Max i3, Molecular devices, San Jose, CA, USA) with excitation and emission wavelengths of 490 nm and 520 nm, respectively. The BACE enzyme and substrate mixture were used as positive controls. The inhibition of BACE activity comparing the rate of reaction of the sample to that of the blank. All experiments were performed in triplicate and the percentage of inhibition of BACE activity was calculated according to the following Equation:(1)$$\mathrm{BACE}\;\mathrm{activity}\;\mathrm{inhibition}\;(\%)\;=\;1-\frac{S-\;S\;\prime }{P-\;P\;\prime }\;\times \;100$$

s: Assay buffer with substrate solution, enzyme with test compound or inhibitor sample

s’: Assay buffer with substrate solution, enzyme without test compound or inhibitor sample

p: Assay buffer with substrate and enzyme solution

p’: Assay buffer with substrate solution without enzyme solution

### 2.8. 2,2′-Azino-Bis-(3-Ethylbenzothiazoline-6-Sulfonic Acid) (ABTS) Radical Scavenging Activity

ABTS radical cations were produced by reacting a 7 mM ABTS solution with 2.45 mM potassium persulfate in the dark at room temperature for 16 h. Absorbance of the reactant was later adjusted to 0.7 at 734 nm. Aliquots of BJIGT solution (100 μL) at various concentrations were mixed with 100 μL ABTS^+^ solution. The reaction mixture was incubated for 5 min in the dark at room temperature. The absorbance of the resulting solution was measured at 734 nm using an Epoch Microplate Spectrophotometer (BioTek Instruments, Winooski, VT, USA). The radical scavenging capacity of the BJIGT-treated samples were calculated using the following Equation:(2)$$\mathrm{Scavenging}\;\mathrm{activity}\;\left(\%\right)\;=\;\frac{1-\mathrm{Absorbance}\;\mathrm{of}\;\mathrm{BJIGT}-\mathrm{treated}\;\mathrm{sample}}{\mathrm{Absorbance}\;\mathrm{of}\;\mathrm{untreated}\;\mathrm{sample}}\;\times \;100$$

### 2.9. 2,2-Diphenyl-1-picrylhydrazyl (DPPH) Radical Scavenging Activity

To determine DPPH radical scavenging activity, a 100 μL aliquot of BJIGT at various concentrations was mixed with 100 μL DPPH solution (0.15 mM in methanol). The reaction mixture was then incubated for 30 min in the dark at room temperature. The absorbance of the resulting solution was measured at 517 nm using an Epoch Microplate Spectrophotometer (BioTek Instruments). The radical scavenging capacity of each tested sample was calculated using the formula above.

### 2.10. Cell Viability Assay

HT22 cells were maintained in Dulbecco’s Modified Eagle’s medium (Hyclone/Thermo, Rockford, IL, USA), supplemented with 10% fetal bovine serum (Hyclone/Thermo, Rockford, IL, USA) and penicillin/streptomycin, in a 5% CO_2_ incubation environment at 37 °C. Cells were plated onto 96-well microplates at a density of 5 x 10^3^/well and treated with various concentrations of each herbal extract for 24 h. Cell Counting Kit (CCK)-8 solution (Dojindo, Kumamoto, Japan) was added, and the cells were incubated for 4 h. The absorbance for each well was read at 450 nm on an Epoch Microplate Spectrophotometer (BioTek Instruments, Inc., Winooski, VT, USA). Cell viability was determined using the following Equation:(3)$$\mathrm{Cell}\;\mathrm{viability}\;\left(\%\right)=\frac{\mathrm{Mean}\;\mathrm{OD}\;\mathrm{in}\;\mathrm{drug}-\mathrm{treated}\;\mathrm{cells}}{\mathrm{Mean}\;\mathrm{OD}\;\mathrm{in}\;\mathrm{untreated}\;\mathrm{cells}}\times 100$$

### 2.11. Animals

Imprinting Control Region (ICR) male mice (8 weeks old) were purchased from Dae Han Biolink (Eumseong, South Korea) and housed in pathogen-free, environmentally-regulated conditions (22 °C, 12 h light/12 h dark cycle). Animals were allowed water and standard food pellets ad libitum. After 1 week of acclimation to the animal care facility, mice were divided into six groups (*n* = 8–10 per group): Normal control, Aβ-injected, Aβ-injection + oral BJIGT gavage (200, 400, or 800 mg/kg/day), and Aβ-injection + oral morin administration (10 mg/kg/day). All animals in all groups were treated for 18 days. All experimental procedures were conducted in accordance with the National Institutes of Health Guidelines for the Care and Use of Laboratory Animals and were approved by the Institutional Animal Care and Use Committee of the Korea Institute of Oriental Medicine (Approval number: 17-044). Animal handling was carried out in accordance with the dictates of the National Animal Welfare Law of Korea.

### 2.12. Intracerebroventricular (ICV) Injection of Aβ_1-42_

The Aβ_1-42_ (Aβ) peptide was purchased from AnaSpec (Fremont, MO, USA) and prepared by dissolving in PBS to 1 mM and incubating at 37 °C for 5 days to encourage the formation of Aβ aggregates. Incubated Aβ (10 μmol/mice) was then infused stereotaxically (anteroposterior = −0.5 mm, mediolateral = 1 mm, and dorsoventral = 2.5 mm) [15]. For this procedure, a Hamilton syringe with a 26-gauge stainless-steel was used to deliver 3 μL to each mouse’s ICV region at a rate of 1 μL/min. Animal body temperatures were maintained at 36.5 ± 0.5 °C. Normal control animals were subjected to the same procedure but treated with vehicle (saline) rather than Aβ.

### 2.13. Passive Avoidance Test

The passive avoidance test was performed in an apparatus consisting of two identically sized chambers (40 cm × 20 cm × 30 cm) separated by a guillotine door (5 cm × 5 cm). One chamber was lit with a 60 W bulb and the other was unlit. The floor of the darkened compartment was comprised of 5 mm stainless steel rods spaced 1 cm apart. Mice were administered either saline or BJIGT daily for 18 days. On day 14, animals were individually placed into the light compartment and allowed to freely explore it. After 20 s, the guillotine door was raised to allow entry into the dark compartment. On day 15, the same procedure was repeated and an electric foot shock (0.5 mA) 3 s in duration was delivered immediately to the animal via the grid floor after it entered the dark compartment. Mice were then tested on day 16, immediately after administration of Aβ. The latency to enter the dark compartment was recorded during a maximum 300 s period.

### 2.14. Y-Maze Test

The Y-maze test was carried out 18 days after Aβ_1-42_ injection. This apparatus was composed of black painted wood and three arms, labeled A, B, or C, each of which was 40 cm long, 12 cm high, 3 cm wide at the bottom, and 10 cm wide at the top. The arms converged in a central equilateral triangle that was 4 cm along each axis. Each mouse was placed in one arm and allowed to move freely through the maze for 8 min. The sequence of arm entries was manually recorded. Alternation was defined as a successive entry into the arms in triplet sets. For example, if the animal first entered A then B then C, this would count as one alternation, while an animal that enters B then A then B would not have been considered to have alternated. The alternation behavior (%) was calculated as: Alternation behavior (%) = (actual alternations)/(possible alternations) × 100

### 2.15. Brain Tissue Preparation

Mice were anesthetized and perfused transcardially with 0.05 mol/L PBS followed by cold 4% PFA in 0.1 mol/L phosphate buffer. Brains were removed and postfixed in 0.1 mol/L phosphate buffer containing 4% PFA overnight at 4 °C and then immersed in a solution containing 30% sucrose in 0.05 mol/L PBS for cryoprotection. Serial 30 μm-thick coronal sections were sectioned via a freezing microtome (Leica Instruments GmbH, Nussloch, Germany) and stored in cryoprotectant (25% ethylene glycol, 25% glycerol, and 0.05 mol/L phosphate buffer) at 4 °C until use. For fresh tissue analyses, mice were decapitated and their brains isolated and stored at −80 °C until use.

### 2.16. Western Blot Analysis

Hippocampus tissues from BJIGT-treated mice were washed three times with PBS and lysed in lysis buffer (1% Triton X-100, 1% deoxycholate, and 0.1% NaN3) containing protease inhibitors (Roche Diagnostics, Mannheim, Germany). Cells were lysed in a CelLytic M lysis buffer (Sigma-Aldrich) containing a protease inhibitor cocktail (GenDEPOT, Barker, TX) to prepare whole cell extracts. The sample’s protein concentration was then determined using a Bradford protein assay reagent (Sigma-Aldrich). Equal amounts of cell extract (20–30 μg) were resolved by 4–20% sodium dodecyl sulfate-polyacrylamide gel electrophoresis (SDS-PAGE) and transferred to a polyvinylidene fluoride (PVDF) membrane. This membrane was then incubated with 5% skim milk in Tris-buffered saline containing Tween 20 (TBST), followed by an overnight incubation at 4 °C with the appropriate primary antibodies. Antibodies used included anti-Aβ (Abcam, Cambridge, MA, USA), anti-BDNF (brain-derived neurotrophic factor), and glyceraldehyde 3-phosphate dehydrogenase (GAPDH) (Santa Cruz Biotechnology, Dallas, TX, USA). The membranes were then washed three times with TBST and incubated with a horseradish peroxidase (HRP)-conjugated secondary antibody (Jackson ImmunoResearch, West Grove, PA, USA) for 1 h at room temperature. The membranes were again washed three times with TBST and immunoreactivity was then visualized using an enhanced chemiluminescence (ECL) kit (Thermo Scientific, Rockford, IL, USA). Images were captured using the ImageQuant LAS 4000 mini Luminescent Image Analyzer (GE Healthcare Bio-Sciences, Piscataway, NJ, USA).

### 2.17. Immunohistochemistry

For immunohistochemical analyses, brain sections were rinsed briefly in PBS and treated with 1% hydrogen peroxide for 15 min. Sections were then incubated with rabbit anti-NeuN (1:1000 dilution; Merck Millipore, Darmstadt, Germany) overnight at 4°C in the presence of 0.3% Triton X-100 and normal goat serum. After rinsing in PBS, the sections were incubated with biotinylated anti-rabbit IgG (1:500) for 1 h, rinsed, and incubated with ABC (1:100) for 1 h at room temperature. Peroxidase activity was visualized by incubating sections with DAB in 0.05 mol/L Tris-buffered saline (TBS, pH 7.6). After several rinses with PBS, sections were mounted on gelatin-coated slides, dehydrated, and coverslipped using histomount medium.

### 2.18. Immunofluorescent Staining

The brain sections were rinsed briefly in PBS and treated with 0.5% BSA for 30 min. The sections were incubated with rabbit anti-Aβ (1:500 dilution; Abcam, Cambridge, UK) overnight at 4 °C in the presence of 0.3% Triton X-100 and normal goat serum. They were then incubated for 2 h with an Alexa Fluor conjugated secondary antibody (diluted 1:500). Finally, sections were washed in PBS and mounted using Vectashield mounting medium containing DAPI (Vector Labs, Burlingame, USA). The images were taken using a fluorescence microscope (Olympus Microscope System CKX53; Olympus, Tokyo, Japan). A threshold for positive staining was determined for each image which included all cell bodies and processes but excluded any background staining.

### 2.19. Statistical Analyses

Results are expressed as the mean ± standard error of measurement. All of the experiments were performed at least 3 times. One-way analyses of variance (ANOVA) tests were used to detect significant differences between the control and treatment groups. All statistical tests were performed using GraphPad Prism 7.0 software (GraphPad Software, San Diego, CA, USA). Dunnett’s test was used for multiple comparisons. Statistical differences were considered significant at *p* < 0.05.

## 3. Results

### 3.1. In Vitro Effects of BJIGT on Aβ Aggregation and BACE Activation

Aβ accumulation is a critical component of the histopathological progression of dementias and particularly of Alzheimer’s disease [16]. Given this, the effect of BJIGT on Aβ aggregation was a point of interest and thus assessed here. BJIGT increased the inhibition of Aβ aggregation in a dose-dependent manner (Figure 1A). We also measured the activity of BACE, an enzyme involved in the generation of Aβ peptides, which forms aggregates in the brains of AD patients [17]. BJIGT increased the inhibition of BACE activity in a dose-dependent manner (Figure 1B). The antioxidant activity of BJIGT was evaluated by measuring radical scavenging activity in the BJIGT- and control-treated tissues, as oxidative stress is one possible cause of dementia [18]. BJIGT dose-dependently increased the radical scavenging activity of ABTS and DPPH (Figure 1C and D, respectively).

### 3.2. Neuroprotective Effect of BJIGT in H_2_O_2_-Damaged HT22 Hippocampal Cells

A cell viability assay, the CCK assay, was performed to determine whether BJIGT is cytotoxic against HT22 cells. HT22 cells were treated with various concentrations (0, 12.5, 25, or 50 μg/mL) of BJIGT for 24 h. BJIGT was found to have no significant effect on the viability of HT22 cells (Figure 1E). Following this, HT22 cells were exposed to H_2_O_2_ to examine the effect of BJIGT on neuronal cell damage. H_2_O_2_ treatment significantly reduced cell viability. By contrast, BJIGT significantly inhibited H_2_O_2_-mediated cell death (Figure 1F).

### 3.3. Ameliorating Effect of BJIGT on Memory Impairment in Aβ-Injected AD-Like Mouse Model

To evaluate the memory-enhancing effects of BJIGT, passive avoidance and Y-maze tests were conducted in an Aβ-injected AD-like mouse model (Figure 2A). An ICV injection of Aβ aggregates significantly shortened the passive avoidance latency in treated animals as compared with controls (*p* < 0.01). By contrast, BJIGT treatment significantly increased this latency compared when compared with animals from the Aβ group (*p* < 0.001) (Figure 2B). In the Y-maze test, BJIGT significantly reversed deficits in the spontaneous alternation behavior of Aβ-injected mice (Figure 2C). No significant difference was observed in the number of arm entries between animals from the two groups created (Figure 2D). Morin was used as a positive control.

### 3.4. Effects of BJIGT on Hippocampal Expression of Neuronal Markers and Aβ Accumulation in an Aβ-Injected Ad-Like Mouse Model

To determine whether Aβ accumulation was affected by the ICV injection of Aβ aggregates in the brain, immunofluorescence was used to visualize Aβ with an anti-Aβ antibody. These studies revealed that injections of Aβ enhanced immunofluorescent reactivity with anti-Aβ in injected brain tissues. In contrast, when Aβ-injected mice were treated with BJIGT they presented with reduced immunofluorescent reactivity in the brain (Figure 3A). BJIGT significantly decreased the relative ratio of Aβ-specific activity and Aβ protein expression in hippocampal tissues compared with controls (Figure 3B and C, respectively).

NeuN, a neuron-specific protein, is a useful neuronal marker [19]. In normal mice, NeuN immunoreactive neurons were clearly observed in the striatum pyramidale (SP) of the hippocampus. In contrast, the Aβ injection suppressed NeuN labeling in the SP of the hippocampus, including in the CA3 region. BJIGT administration markedly reversed this lowered NeuN immunoreactivity (Figure 4A).

BDNF is another neuronal marker that plays an important role in the promotion of neuronal cell survival [20]. Aβ administration markedly reduced the expression of BDNF (Figure 4B). In contrast, BJIGT treatment blocked the Aβ-mediated suppression of BDNF in a dose-dependent manner. Morin also reversed the decreased expression of BDNF in Aβ-injected mice.

### 3.5. Optimization of HPLC Separation

We used HPLC to separate the seven standard components of BJIGT (Figure 5). We obtained good separation chromatograms using mobile phases consisting of 1.0% (v/v) aqueous acetic acid (A) and acetonitrile (B). The ultraviolet (UV) wavelengths for detecting these multiple components of BJIGT were 254 nm for glycyrrhizin; 280 nm for liquiritin apioside, liquiritin, and hesperidin; and 330 nm for nodakenin, decursin, and decursinol angelate. Using established HPLC methods, the seven standard components were resolved within 45 min. The retention times of the liquiritin apioside, liquiritin, nodakenin, hesperidin, glycyrrhizin, decursin, and decursinol angelate were 12.65, 13.18, 13.67, 15.38, 33.85, 43.21, and 43.61 min, respectively. HPLC chromatograms of the formulation of Bojungikgi-tang and the standard mixture are shown in Figure 5.

### 3.6. Linearity, LOD, and LOQ

The linear relationships between the peak areas (*y*) and concentrations (*x*, μg/mL) of the components of BJIGT were expressed by regression equations (*y* = a*x* + b) (Table 1). The calibration curves for these seven components demonstrated good linearity (*r*^2^ ≥ 0.9997). The LODs and LOQs for the tested components were 0.052–0.797 μg/mL and 0.157–2.415 μg/mL, respectively.

### 3.7. Determination of the Seven Standard Components of BJIGT

Established HPLC analytical methods were applied to the simultaneous quantification of the seven components of BJIGT. These standard components ranged in quantity from 0.109 to 1.766 mg/g (for all quantities, refer to Table 2). Among these seven components, glycyrrhizin was the most abundant compound in BJIGT.

## 4. Discussion

Dementias are neurodegenerative diseases that result in a progressive and irreversible loss of neurons and brain functioning. According to the 2016 World Alzheimer Report, approximately 47 million individuals worldwide have dementia, a figure which is predicted to reach over 131 million by 2050 [21]. Given this predicted rise, the cost of medical care for dementia patients will be a serious future problem. The total worldwide cost of this care is estimated to reach one trillion US dollars by 2018. Unfortunately, there are no cures for any of the dementias including AD, VD, Lewy body dementia, frontotemporal dementia, or mixed dementia (a type of dementia where the patient experiences two or more disorders, at least one of which is dementia). AD is the most common type of dementia and is a life-threatening disease. There are three major hypotheses regarding the biochemistry that underlies AD. These include the cholinergic hypothesis, the Aβ hypothesis, and the Tau hypothesis [22]. At present, most available treatments for AD, including tacrine, rivastigmine, galantamine, and donepezil, target acetylcholinesterase, a key nervous system enzyme [23]. However, these medications function only to relieve some dementia symptoms or to delay its progression, and do not cure the underlying disease. Amyloid plaque formation in the brain is one hallmark endophenotype of AD [24]. Thus, many recent AD drug development studies have focused on the Aβ hypothesis, though a debate continues with respect to the value of Aβ as an AD drug target.

In the present study, we report that BJIGT, a traditional herbal formula, has multiple anti-AD effects in AD-like experimental models. An in vitro Aβ aggregation assay revealed that BJIGT inhibited the aggregation of Aβ in a dose-dependent manner. In an Aβ-injected AD mouse model, in which Aβ accumulation in the brain was observed using immunofluorescence, BJIGT administration reduced Aβ accumulation when compared with untreated controls. Moreover, Aβ aggregation in hippocampus was significantly decreased with BJIGT administration, consistent with results of the in vitro assay.

Aβ aggregation is generated by cleavage of the amyloid precursor protein (APP) by BACE, an aspartic acid protease [25]. BACE is considered to be a promising new therapeutic target for AD treatment. BJIGT extract increased the inhibition of BACE activity in a dose-dependent manner, consistent with the inhibition of Aβ aggregation by BJIGT also reported here. These results indicate that BJIGT potentially acts as an inhibitor of Aβ aggregation.

Oxidative stress accumulation, which is detectable in even the earliest stages of AD progression, plays a role in AD pathogenesis [26]. Accumulating evidence suggests that elevated levels of Aβ increase levels of oxidation products in the hippocampus and cortex of AD patients [27,28,29]. Moreover, the abnormal aggregation of Aβ and the deposition of neurofibrillary tangles in the brains of AD patients interact with oxidative stress to induce further neuronal cell damage [30]. Thus, consideration of antioxidant activity and mechanisms is essential to the identification of additional AD treatment candidates. Natural products, including herbal medicines, have the potential to target oxidative stress mechanisms. Numerous reports have highlighted the antioxidant activity of natural products in AD-like experimental models [31]. To examine the antioxidant effect of BJIGT, one such natural product, we evaluated the scavenging of free radicals including ABTS and DPPH. We report here that BJIGT markedly increased scavenging of ABTS and DPPH, implying that BJIGT acts as an antioxidant. Additionally, we report that BJIGT significantly prevents neuronal cell death, a major element of the progression of AD. BJIGT treatment significantly ameliorated H_2_O_2_-induced cell death in HT22 hippocampal cells, suggesting that BJIGT may act as a neuroprotective agent.

One of AD’s earliest and most central symptoms is memory loss, which leads to a disruption of social activities [32]. The hippocampus is a key brain region that is responsible for the regulation of memory systems. Memory loss in AD patients is thought to result from changes to the hippocampus and its fiber tracts, collectively known as the hippocampal formation [33]. Given this, we assessed passive avoidance and Y-maze behavior in an AD-like mouse model to investigate whether BJIGT might have an ameliorating effect on deficits to these memory-dependent tests. Memory deficits were induced in this model by acute ICV injection of Aβ aggregates into the brain. In Aβ-injected mice, we observed a significant reduction in the passive avoidance latency and the spontaneous alternation or working memory (Y-maze) behaviors, evidencing memory deficits in these animals. BJIGT administration significantly inhibited the development of these memory impairments after Aβ injection, suggesting that BJIGT rescues Aβ-dependent cognitive deficits in a mouse model. In addition, BJIGT administration reversed NeuN and BDNF expression deficits in the hippocampi of Aβ-injected mice. Taken together, our in vitro and in vivo results support the conclusion that BJIGT has great potential to act as an anti-AD agent.

BJIGT, which is composed of eight herbal medicines (Table 3) including Astragali Radix, Ginseng Radix, Atractylodis Rhizoma Alba, Glycyrrhizae Radix et Rhizoma, Angelicae Gigantis Radix, Citri Unshius Pericarpium, Cimicifugae Rhizoma, and Bupleuri Radix, has been identified as an effective agent for the treatment of acute gastric injury [34], allergic rhinitis [35], and fatigue associated with chronic diseases [36]. The main classes of ingredients in Bojungikgi-tang are as follows: Flavonoids (e.g., formononetin and calycosin) and triterpene saponins (astragaloside I–IV) from Astragali Radix [37]; triterpene saponins (e.g., ginsenoside Rg1 and Rb2) from Ginseng Radix [38]; sesquiterpenes (e.g., atractylon and atractylenolide Ⅰ–Ⅲ) from Atractylodis Rhizoma Alba [39]; flavonoids (e.g., liquiritin and liquiritigenin) and triterpene saponins (e.g., glycyrrhizin) from Glycyrrhizae Radix et Rhizoma [40]; coumarins (e.g., decursin, decursinol angelate, and nodakenin) from Angelicae Gigantis Radix [41]; flavonoids (e.g., hesperidin) from Citri Unshius Pericarpium [42]; phenols (e.g., ferulic acid and isoferulic acid) from Cimicifugae Rhizoma [43]; and triterpene saponins (e.g., saikosaponin A, C, and D) from Bupleuri Radix [44]. We were able to determine the following seven components of the Bojungikgi-tang formulation used in the present study via HPLC-PDA: liquiritin apioside, liquiritin, nodakenin, hesperidin, glycyrrhizin, decursin, and decursinol angelate. Of these, glycyrrhizin (1.766 mg/g), a marker compound for Glycyrrhizae Radix, was the most abundant.

Several studies have examined the anti-AD effects of individual BJIGT component compounds. Liquiritin, for example, has been reported to exhibit neuroprotective and neurotrophic effects on primary cultured hippocampal cells and may also relieve pain and cough symptoms, as well as prevent inflammation [45,46,47,48]. Hesperidin has been reported to downregulate autophagy in insulin-stimulated neuronal cells, which may be one of the mechanisms by which it further corrects the energy metabolism impairments known to lead to neuronal injury in early AD [49]. Song et al. reported that glycyrrhizin effectively ameliorated memory deficits induced by systemic LPS treatment via the inhibition of pro-inflammatory cytokines and microglial activation in the brain [49]. Decursin and decursinol angelate have been reported to stimulate the activation of Nrf2 in PC12 cells, thereby elevating the activity of other, antioxidant enzymes that contribute to defensive mechanisms which work against Aβ-induced oxidative injury [50]. Taken together, these reports and the data presented here suggest that the anti-AD effect of BJIGT results from a synergistic interaction between multiple BJIGT compound components.

## 5. Conclusions

In the present study, we provide scientific evidence to support the conclusion that BJIGT may successfully be used in the treatment of AD and AD-related diseases. We demonstrated that BJIGT inhibited amyloid-β (Aβ) aggregation and BACE activity, as well as increased antioxidant activity. BJIGT was further revealed to have protective effects against neuronal damage in a hippocampal cell line. In addition, BJIGT administration improved memory impairments and reversed decrements to the expression of the neuronal marker NeuN and neurotrophic growth factor BDNF in an AD-like mouse model. Collectively, our findings suggest that BJIGT may serve as a future therapeutic agent for the treatment of AD.

## Figures and Tables

**Figure 1 nutrients-10-01952-f001:**
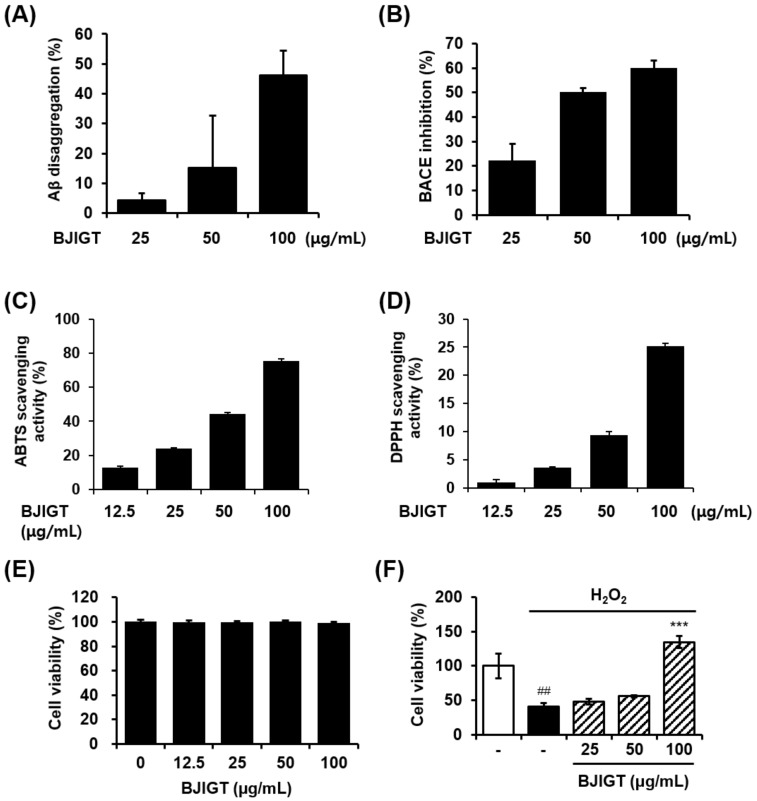
In vitro activity of Bojunikgi-tang (BJIGT) on Alzheimer’s disease (AD)-associated biomarkers. (**A**) The effect of BJIGT on Aβ aggregation was evaluated. Aβ aggregates were treated with BJIGT extracts (0, 25, 50, 100 μg/mL), reacted with aggregates of Aβ_1-42_ peptides and 2 mM thioflavin T for 2 h at 37 °C. Fluorescence intensity was measured at Ex/Em = 440/ 480 nm. (**B**) The effect of BJIGT on β-secretase (BACE) activity was evaluated. The BACE activity was treated with BJIGT extracts (0, 25, 50, 100 μg/mL), and reacted with BACE substrate solution. Fluorescence intensity was measured at Ex/Em = 490/ 520 nm. (**C** and **D**) Antioxidant activity of BJIGT was assessed by a free radical scavenging assay for ABTS (**C**) and DPPH (**D**). (**E**) Cytotoxicity of BJIGT in HT22 hippocampal cells were measured using the Cell Counting Kit (CCK) assay. Cells were treated with various concentrations of BJIGT for 24 h. The results of three independent experiments are expressed as mean ± SEM. (**F**) HT22 cells were exposed to H_2_O_2_ in the absence or presence or BJIGT for 6 h. Cell viability was determined using the CCK assay. The data are presented as means ± S.E.M. ^##^
*p* < 0.01 vs. normal group, ^***^
*p* < 0.001 vs. H_2_O_2_-treated cells via two-tailed Student’s *t*-test.

**Figure 2 nutrients-10-01952-f002:**
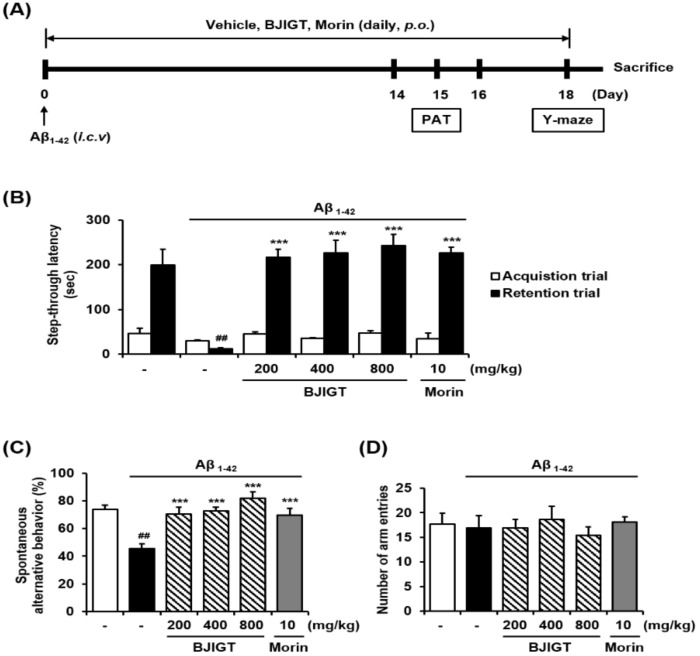
Effect of BJIGT on behavioral memory impairments in an Aβ-induced AD mouse model. (**A**) Timeline of the animal experiments. (**B**) Male imprinting control region (ICR) mice were assigned to six groups (*n* = 7/group). Aβ aggregates (50 pmol per 10% DMSO in PBS) were intracerebroventricular (ICV) injected with vehicle (saline) or BJIGT (200, 400, 800 mg/kg) or administered morin (10 mg/kg), an inhibitor of Aβ aggregation used as a positive control, orally for 18 days after one week of acclimation. (**B**) For the passive avoidance test, mice were trained on a one-trial step-through passive avoidance task on day 14 after Aβ injection (retention trial). The test trial was administered 2 days after the training trial (acquisition trial). The latency time was recorded in both retention and acquisition trials. (**C** and **D**) The Y-maze was performed on day 18 after Aβ injection. Spontaneous alternation behavior (**C**) and total arm entries (**D**) were measured across an 8-min session. Results are presented as means ± S.E.M. ^##^
*p* < 0.01 vs. normal group, ^***^
*p* < 0.001 vs. Aβ control group via a two-tailed Student’s *t*-test.

**Figure 3 nutrients-10-01952-f003:**
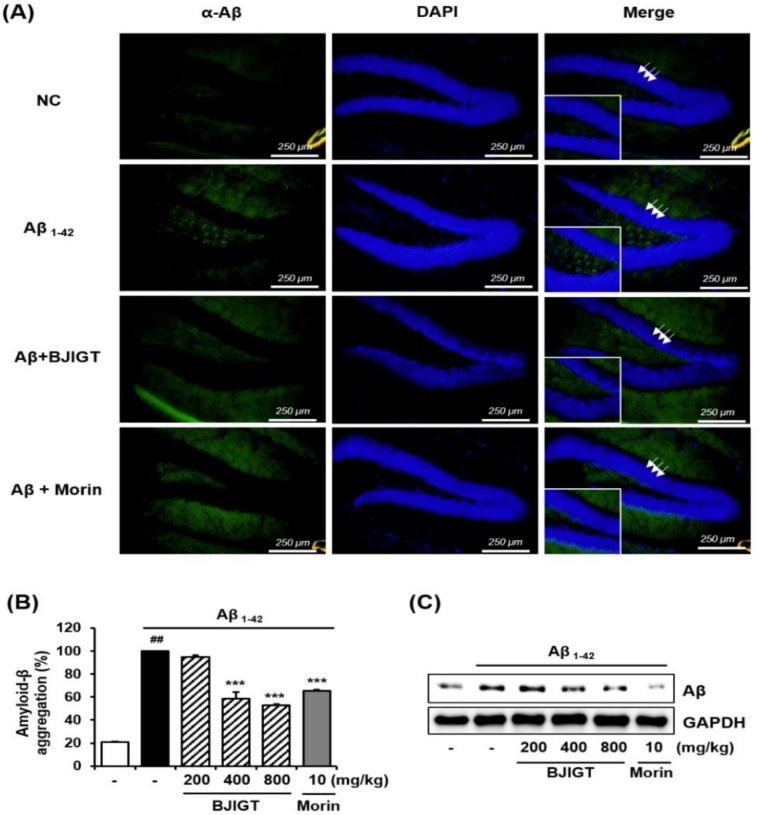
Effects of BJIGT on Aβ aggregation in an Aβ-induced AD mouse model. (**A**) Immunofluorescence was used to evaluate the effect of BJIGT on Aβ aggregation in the hippocampus of experimental mouse. Antibody labeling via anti-Aβ was used across multiple conditions: NC, normal control; Aβ, Aβ-injected AD control; Aβ + morin, injection of Aβ and administration of 10 mg/kg/day of morin; and Aβ + BJIGT 800, injection of Aβ and administration of 800 mg/kg/day of BJIGT. Morin, which inhibits Aβ aggregation, was used as a positive control. (**B**) To assess Aβ aggregation, total hippocampal proteins were extracted and reacted with aggregates of Aβ_1-42_ peptides and 2 mM thioflavin T for 2 h at 37 °C. Fluorescence intensity was measured at Ex/Em=440/ 480 nm. The effect of BJIGT on Aβ aggregation was expressed as a percentage relative to aggregation in the Aβ-injected group. Results are presented as means ± S.E.M. ^##^
*p* < 0.01 vs. normal group, ^***^
*p* < 0.001 vs. Aβ control group via a two-tailed Student’s *t*-test. (**C**) Total hippocampal protein was prepared and western blotting was used to detect levels of Aβ. Glyceraldehyde 3-phosphate dehydrogenase (GAPDH) was used as an internal control.

**Figure 4 nutrients-10-01952-f004:**
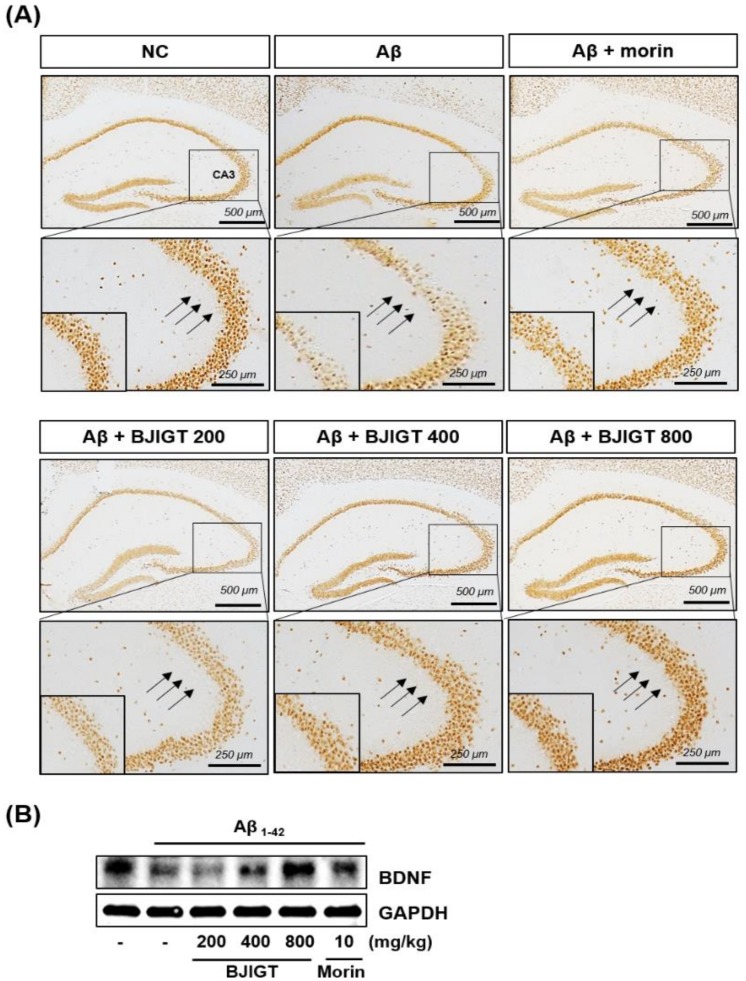
The effect of BJIGT on expression of neuronal marker NeuN and neurotrophic growth factor (brain-derived neurotrophic factor; BDNF) in an Aβ-induced AD mouse model. (**A**) Hippocampal expression of NeuN was determined by immunohistochemistry using anti-NeuN (magnification, ×200) across multiple conditions: NC, normal control; Aβ, Aβ-injected AD control; Aβ + morin, injection of Aβ and administration of 10 mg/kg/day of morin; and Aβ + BJIGT 200, 400 or 800, injection of Aβ and administration of 200, 400 or 800 mg/kg/day of BJIGT. (**B**) Total hippocampal protein, prepared and subjected to western blotting, was used to detect levels of BDNF. GAPDH was used as an internal control.

**Figure 5 nutrients-10-01952-f005:**
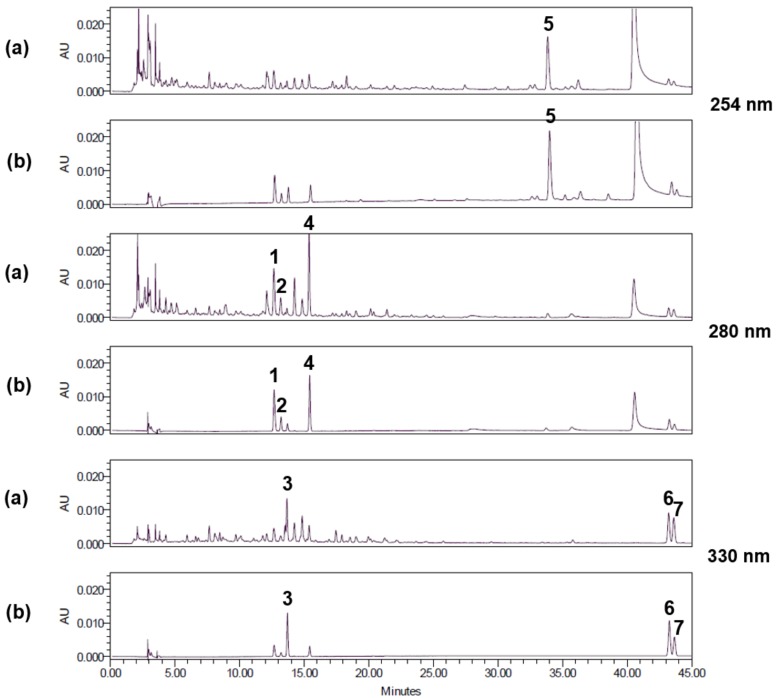
High-performance liquid chromatography (HPLC) chromatograms of BJIGT components (**a**) and a standard mixture (**b**) at 254 nm, 280 nm, and 330 nm. Components included: Liquiritin apioside (**1**), liquiritin (**2**), nodakenin (**3**), hesperidin (**4**), glycyrrhizin (**5)**, decursin (**6**), decursinol angelate (**7**).

**Table 1 nutrients-10-01952-t001:** Linear range, regression equation, correlation coefficients, LODs, and LOQs for ompounds.

Compound	Linear Range (μg/mL)	Regression Equation (y = ax+b) ^a^		Correlation Coefficient (*r*^2^)	LOD ^b^ (μg/mL)	LOQ ^c^ (μg/mL)
		Slope (a)	Intercept (b)			
Liquiritin apioside	1.56–100	14986	3467.2	1.0000	0.228	0.692
Liquiritin	0.39–25	18335	1582.3	1.0000	0.052	0.157
Nodakenin	0.78–50	24593	3093.5	0.9999	0.189	0.574
Hesperidin	1.56–100	17953	1857.5	0.9999	0.441	1.336
Glycyrrhizin	6.25–400	5195.6	1474.1	1.0000	0.797	2.415
Decursin	0.78–50	32417	10204	0.9997	0.221	0.669
Decursinol angelate	0.39–25	35549	4903.6	0.9997	0.108	0.328

^a^ y = ax + b, y means peak area and x means concentration (μg /mL); ^b^ LOD (Limit of detection): 3.3 × (SD of the response / slope of the calibration curve); ^c^ LOQ (Limit of quantitation): 10 × (SD of the response / slope of the calibration curve).

**Table 2 nutrients-10-01952-t002:** The content of standard compounds in Bojungikgi-tang.

Compound	Content (mg/g)
Liquiritin apioside	0.400 ± 0.001
Liquiritin	0.110 ± 0.000
Nodakenin	0.158 ± 0.000
Hesperidin	0.487 ± 0.001
Glycyrrhizin	1.766 ± 0.003
Decursin	0.136 ± 0.000
Decursinol angelate	0.109 ± 0.001

**Table 3 nutrients-10-01952-t003:** Composition of Bojungikgi-tang.

Herbal Medicine	Scientific Name of Plant Source	Family	Origin
Astragali Radix	*Astragalus membranaceus* Bunge	Fabaceae	South Korea
Ginseng Radix	*Panax ginseng* C. A. Meyer	Araliaceae	South Korea
Atractylodis Rhizoma Alba	*Atractylodes japonica* Koidz.	Compositae	China
Glycyrrhizae Radix et Rhizoma	*Glycyrrhiza uralensis* Fischer	Fabaceae	China
Angelicae Gigantis Radix	*Angelica gigas* Nakai	Apiaceae	South Korea
Citri Unshius Pericarpium	*Citrus reticulata* Blanco	Rutaceae	South Korea
Cimicifugae Rhizoma	*Cimicifuga heracleifolia* Kom.	Ranunculaceae	China
Bupleuri Radix	*Bupleurum falcatum* L.	Apiaceae	China

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
