# Peer review of "Bojungikgi-Tang, a Traditional Herbal Formula, Exerts Neuroprotective Effects and Ameliorates Memory Impairments in Alzheimer’s Disease-Like Experimental Models"

_nutrients, 2018, doi:10.3390/nu10121952_

Round 1
Reviewer 1 Report
Why only male mice were selected?
Please include if has been described adverse effects for BJIGT intake
Discussion should be extended i.e. including how the cited actives have demonstrated efficacy in previous works. This include the extract tested, if in-vivo or in-vitro essays has been selected, the dose administered, the extract prepared (if applied) ...
Reviewer 2 Report
The article is adequately organized and well written.
However, the plant familes mentioned in Table 3 should be revised, as you place Astragalus in the Leguminosae and Glycyrrhiza in the Fabaceae. For sake of uniformity, both should be placed under the Fabaceae.
Also, the genus Citrus is in the Rutaceae, not Oleaceae family.
In the main text on page 14, the sentence after reference [23] mentions the the word "relief" , which should read: relieve.
Author Response
Comments and Suggestions for Authors
The article is adequately organized and well written.
However, the plant familes mentioned in Table 3 should be revised, as you place Astragalus in the Leguminosae and Glycyrrhiza in the Fabaceae. For sake of uniformity, both should be placed under the Fabaceae. Also, the genus Citrus is in the Rutaceae, not Oleaceae family.
à We appreciate your comment. We corrected the family in table 3 as reviewer’s comment.
In the main text on page 14, the sentence after reference [23] mentions the the word "relief" , which should read: relieve.
à We corrected “relief” to “relieve” as reviewer’s comment.

Reviewer 3 Report
The manuscript is well constructed and described in vitro & in vivo inhibition of Aβ through Bojungikgi-tang herbal preparation. The introduction, materials and methods as well as the results are presented clearly with the excellent figures put them clearly into content. My major concern with quantified extract bioactive constituents which should be describe in accordance to their structure activity relationship against each assays and inhibition.
Author Response
Comments and Suggestions for Authors
The manuscript is well constructed and described in vitro & in vivo inhibition of Aβ through Bojungikgi-tang herbal preparation. The introduction, materials and methods as well as the results are presented clearly with the excellent figures put them clearly into content. My major concern with quantified extract bioactive constituents which should be describe in accordance to their structure activity relationship against each assays and inhibition.
à We sincerely agree with the reviewers' comments. Medicinal plants contains hundreds if not thousands of unique compounds. Models based on single molecular entities do not accurately describe or capture the complexity of interactions among the constituents in medicine plants and multiconstituent extracts made from them. In the mainstream pharmaceutical view, there is a single active constituent in a plant that explains its activity and which can be isolated and used as a conventional drug. However, this still does not prove that the single compounds are superior to complex mixtures, particularly when possible beneficial effects of various compounds that indirectly support the main desired action have not been assessed. This synergistic effect may be due to the joint action of the compounds contained in the extract, and not to a particular compound. In the present study, we provide scientific evidence to support the conclusion that Bojungikgi-tang may successfully be used in the treatment of AD and AD-related diseases. We also believe that a systematic and active search of the main constituents of Bojungikgi-tang is necessary, and we will conduct an experiment later on for our next manuscript.
